# Diversity and Multiple Infections of *Bartonella* in Red Deer and Deer Keds

**DOI:** 10.3390/pathogens14010006

**Published:** 2024-12-27

**Authors:** Iva Hammerbauerová, Eva Richtrová, Kateřina Kybicová, Petr Pajer, Jan Votýpka

**Affiliations:** 1Department of Parasitology, Faculty of Science, Charles University, 128 00 Prague, Czech Republic; jan.votypka@natur.cuni.cz; 2National Institute of Public Health, Šrobárova 49/48, 100 00 Prague, Czech Republickaterina.kybicova@szu.cz (K.K.); 3Military Health Institute, Military Medical Agency, Tychonova 1, 160 01 Prague, Czech Republic

**Keywords:** *Bartonella*, Lipoptena, deer, zoonosis

## Abstract

Bartonellae are zoonotic pathogens with a broad range of reservoir hosts and vectors. To examine sylvatic *Bartonella* reservoirs, tissue samples of red deer (*Cervus elaphus*, *n* = 114) and their associated deer keds (*Lipoptena cervi*, *n* = 50; *L. fortisetosa*, *n* = 272) collected in the Czech Republic were tested for the presence of *Bartonella* using PCR at four loci (*gltA*, *rpoB*, *nuoG*, ITS); PCR sensitivity was increased significantly by using primers modified for the detection of wildlife-associated bartonellae. One-third of the deer and 70% of the deer keds were *Bartonella* positive; within the tested animal tissues, usually the spleen was positive. The most prevalent *Bartonella* represents an undescribed species related to isolates from Japanese sika deer and *L. fortisetosa*. Additionally, *B. schoenbuchensis* sensu lato and *B. bovis* were found, together making up 17 genotypes characterized by multi-locus sequence typing, all unique compared to previously published sequences. Nanopore sequencing of selected samples revealed an additional 14 unique *Bartonella* genotypes, with up to six genotypes co-infecting one deer, highlighting the diversity of ruminant *Bartonella*. The high COI variety of examined *L*. *cervi* and *L*. *fortisetosa* suggests *L*. *fortisetosa* in central Europe is not a homogenous invasive population.

## 1. Introduction

Bartonellae are alpha-proteobacteria that can cause bacteremia, fever, and other symptoms in mammals, including humans [1]. Apart from the feline *B. henselae*, causing cat scratch disease [2], and the anthroponotic, louse-borne *B. quintana*, causing trench fever [3], 15 other *Bartonella* species associated with diverse animal hosts and arthropod vectors, are known to cause human disease [4,5,6,7,8,9,10,11,12,13,14,15,16,17,18,19,20]. Because of the wide range of domestic and sylvatic reservoir hosts, other social groups besides cat owners, e.g., game hunters, can be exposed to *Bartonella* infection, and a One Health approach is required to understand the risks.

In this study, red deer (*Cervus elaphus*) and their associated deer keds (*Lipoptena* spp.) were examined as potential reservoirs of *Bartonella* spp. Red deer are common in the forests of central Europe and constitute a significant portion of hunted game; in the Czech Republic, roughly 30,000 red deer are hunted per year [21]. Deer-associated zoonoses might pose a risk to hunters who come into close contact with the animals’ blood and innards. Hunting is popular among Czech citizens; there are almost 90,000 registered hunters [22], comprising about a 10th of the Czech population [23]. Besides forestry workers, many other people can encounter deer keds when visiting forests, e.g., during mushroom hunting, a hobby regularly enjoyed by nearly a third of the adult population [24].

*Bartonella* species infecting ruminants and their ectoparasites form a monophyletic branch referred to as lineage 2 [25] with several described species: *B. bovis*, *B. capreoli*, *B. chomelii*, and *B. schoenbuchensis* [26]. Two other species, *B. dromedarii* [27] and *B. melophagi* [14], have been suggested but not validly published under the International Code of Nomenclature of Prokaryotes. However, based on >95% average nucleotide identity (ANI) between *B. chomelii*, *B. capreoli*, *B. melophagi*, and *B. schoenbuchensis*, it has been proposed that they are not separate species but merely genotypes or subspecies of *B. schoenbuchensis* [28]. Ruminant bartonellae can be zoonotic, and *B. schoenbuchensis* and *B. melophagi* have been isolated from the blood of symptomatic patients [14,20]. However, the epidemiological importance of these species is unclear, as bartonellosis is usually diagnosed exclusively by serological methods. Without culture or sequencing, serological tests cannot be used to determine species. In addition, most diagnostic kits target only *B. henselae* and *B. quintana*, so antibodies against species with significantly different antigenic profiles may not be detected at all.

A study investigating different deer species for *Bartognella* found lineage 2 bartonellae in 24 red deer and one white-lipped deer [29]. Another study found *B. schoenbuchensis* in one red deer sample [30]. In other cervids, *B. schoenbuchensis* has been detected in roe deer and moose [31,32,33]. Additionally, the usually cattle-bound *B. bovis* has been found in moose [32].

Two species of deer keds are present in the Czech Republic. *Lipoptena cervi* is native to Europe, while *Lipoptena fortisetosa*, first captured in Europe in 1933, is sometimes considered to be expanding from East Asia [34]. There is ongoing controversy about whether deer keds bite humans or if the skin irritation sometimes observed after contact with keds is due to scratches from their hooked feet and contamination of the wound. Deer ked dermatitis, a pruritic skin condition lasting up to a year, has been reported following contact with deer keds [35]. The symptoms have been attributed to a hypersensitive immune reaction [35,36]; however, some authors have speculated on a *B. schoenbuchensis* etiology [37,38].

Most data on *Bartonella* in cervids come from detection studies conducted on their ectoparasites. *Bartonella schoenbuchensis* sensu lato has been detected in *L. cervi* collected on red deer, roe deer, and moose [37,39,40,41,42,43,44] and *L. fortisetosa* obtained from red deer, roe deer, and sika deer [45,46,47,48]. The presence of *Bartonella* DNA in *L. cervi* pupae and winged individuals (before they suck blood) suggests that vertical transmission is likely [32,39]. *Bartonella* has been shown to proliferate in the gut of *L. cervi* and *L. fortisetosa*, and these keds are assumed to be natural vectors [37,47].

Despite the high prevalence in deer keds, *Bartonella* in deer in Europe has not yet been extensively studied. This study also aimed to improve detection methods for ruminant *Bartonella* and compare the detection rate of different deer tissues. Additionally, the population structure of *Lipoptena* deer keds was examined for clues to the origin of *L. fortisetosa* in Europe.

## 2. Materials and Methods

### 2.1. Sample Collection

Tissue samples and attached keds (wingless) were collected from deer in the Krkonoše National Park, Czech Republic (360 km^2^), during the hunting season (August–October) in 2016, 2020, 2021, and 2022 by hunters. Ear (4 cm^2^) and spleen (8 cm^3^) tissue samples were taken from each deer from season 2020 onwards for DNA extraction; additionally, a cardiac blood sample (1–2 mL, mixed with EDTA to prevent coagulation) was obtained from each deer shot in the 2022 season. Keds were collected from each animal and identified as *L. cervi* or *L. fortisetosa*. All samples were frozen after collection and transported on ice to the laboratory, where they were stored at −20 °C until DNA isolation.

### 2.2. Bartonella Cultivation

Cultivation of *Bartonella* was attempted following the recommended methodology [49]. Aliquots of blood samples after two freeze cycles were mixed with a medium based on Schneider’s Insect Medium (Sigma Aldrich, St. Louis, MO, USA) [50], inoculated onto chocolate agar plates (LabMedia, Jaroměř, Czech Republic), and incubated at 37 °C with 5% CO_2_ for 4 weeks.

### 2.3. DNA Extraction

Deer keds were transferred into Bead Tubes Type D (Macherey-Nagel, Dűren, Germany) and homogenized using MagNA Lyser Instrument (Roche, Basel, Switzerland). Spleen and ear samples (after hair removal) were incubated in lysis buffer (Qiagen, Hilden, Germany) until tissues were completely dissolved. Genomic DNA from keds from 2016 was isolated using the DNeasy tissue kit (Qiagen, Dusseldorf, Germany) and from all other samples using the croBEE NA16 Nucleic Acid Extraction System (GeneProof, Brno, Czech Republic) according to manufacturing protocol. Two aliquots of genomic DNA were extracted from each blood sample. DNA was stored at −20 °C until analysis. The Euler chart showing positive samples was generated with the eulerr package [51,52].

### 2.4. PCR and Sequencing

A random set of 34 *L. fortisetosa* samples and one *L. cervi* sample were genotyped using PCR targeting the COI (cytochrome oxidase subunit I) locus. Additionally, ten *L. fortisetosa* and three *L. cervi* keds from three other locations in the Czech Republic were included to broaden the sample set. The PCR procedure followed the previous description [53].

All samples underwent pre-screening for *Bartonella* using PCR targeting a 379 bp fragment of the *gltA* (citrate synthase) gene. In cases where the amplicon after PCR was weak, the reaction was repeated with the PCR product as the template. The *gltA* sequences obtained via Sanger sequencing were categorized into genotypes based on single-nucleotide polymorphisms (SNPs). Sequences showing double peaks on chromatograms, indicating multiple infections, were excluded from the analysis. From each genotype, a representative sample was selected for full multi-locus sequence typing (MSLT). To ensure this approach is reliable, 17 samples of the same genotype were analyzed by MSLT to test reproducibility.

Four loci were utilized for MLST: *gltA* (a longer fragment of 740 bp), *rpoB* (β subunit of bacterial RNA polymerase), ITS (16S–23S internal transcribed spacer), and *nuoG* (NuoG subunit of type I NADH dehydrogenase). The reactions were performed in the T100 Thermal Cycler (Biorad, Hercules, CA, USA) using the PPP Master Mix (Top-Bio, Vestec, Czech Republic). Due to a high frequency of PCR failure, slightly modified or completely new primers were used for the *gltA*, *rpoB*, and ITS loci based on an in silico analysis of primer annealing sites (see Table 1, Appendix A). For all PCR reactions, a positive and negative control was used. Products of PCR were sequenced by Sanger sequencing at the Charles University facility in BIOCEV. Primer annealing sites were cropped before phylogenetic analysis.

### 2.5. Nanopore Sequencing

A set of 26 DNA samples from deer tissue was selected for Oxford Nanopore Technology (ONT) sequencing of the 740 bp *gltA* PCR products to check for multiple *Bartonella* infections. PCR products were purified and used as a template for the preparation of libraries for ONT sequencing, using ligation sequencing (SQK-LSK109) and native barcoding expansion 1–12 and 13–24 (EXP-NBD104 and EXP-NBD114) kits (Oxford Nanopore Technologies, Oxford, UK) according to the manufacturer´s instructions. Libraries were sequenced on the ONT GridION platform using R9.4 chemistry (Flow-Cell). Base-calling to biologically relevant bases was performed using Guppy v.5.1.13 (Oxford Nanopore Technologies, Oxford, UK) [60]. The sequencing data were processed with the Porechop v.0.2.4 pipeline to trim the barcodes and to discard possible technical chimeric reads, i.e., reads with barcodes or any other technical sequence in the middle of the read [61]. Then, the NanoCLUST pipeline was used to resolve representative sequences.

### 2.6. Phylogenetic Analysis

For the *Lipoptena* phylogenetic analysis, all available *L. cervi* and *L. fortisetosa* sequences from GenBank and the Barcode of Life Data System (BOLD) with listed locations were obtained. In cases where sequences had identical residues and locations, only the longest one was included.

To build the *Bartonella* phylogenetic trees, all sequences of the given locus marked as *B. bovis*, *B. capreoli*, *B. chomelii*, *B. schoenbuchensis*, *B. dromedarii*, *B. melophagi*, *Bartonella* sp. Negev, and *Bartonella* sp. Honshu in GenBank were used. Additionally, the 10 closest BLAST results for each sequence obtained in this study were included. This yielded 181 *gltA*, 123 ITS, 34 *nuoG*, and 160 *rpoB* sequences. An additional seven *gltA*, five *nuoG*, and five *rpoB* sequences were obtained on request from the authors of a recent study in Germany [44]. Identical sequences were excluded, with the longest preferred, and all sequences were cropped to the length of the loci examined in this study. Sequences for concatenated alignment were obtained from annotated whole genome sequences of lineage 2 bartonellae available in GenBank (AGWB01000000, AGWC01000000, CM001844, CP019781, HG977196, MUBG01000000, NC_008783, NZ_AIMA01000000, NZ_CACVBI010000000, NZ_CADDYD010000000, NZ_CADDZX010000000, NZ_CP019789, NZ_CP154603, NZ_JACJIR010000000, NZ_JBCAUK010000000, NZ_JBCAUL010000000, NZ_MWVG01000000).

All phylogenetic analysis was performed in the Geneious Prime program, version 2023.2.1 (Biomatters, Auckland, New Zealand) [62]. Alignments for the phylogenetic trees were obtained using the Geneious alignment algorithm with default settings for the *gltA*, *rpoB*, and *nuoG* loci. For ITS and the concatenated sequences, due to frequent indels, MAFFT alignment was used with the E-INS-i algorithm [63,64]. Phylogenetic trees were created with the MrBayes plugin [65], and bootstrap values (500 replicates) were calculated with the PhyML plugin [66].

The genotype network was created in PopART (version 1.7) with the Minimum Spanning algorithm [67,68].

## 3. Results

### 3.1. Collection of Samples

The number of samples collected is summarized in Table 2. *Lipoptena fortisetosa* was more abundant, with 272 individuals as opposed to 50 individuals of *L. cervi*.

### 3.2. Results of Bartonella Cultivation

Cultivation of *Bartonella* from 10 blood samples from 2022 was unsuccessful, in part due to contaminated blood samples.

### 3.3. Population Structure of Deer Keds

In the analyzed Czech deer keds, six haplotypes of *Lipoptena fortisetosa* and four haplotypes of *Lipoptena cervi* were identified, with both exhibiting similar levels of intra-species diversity. While Europe and northern continental Asia shared similar haplotypes of *L. fortisetosa*, a specimen from Japan and two specimens from Thailand formed distinct branches (Appendix A).

### 3.4. Bartonella Prevalence

Out of the 114 deer examined, 38 (33.3%) tested positive for *Bartonella* DNA in one or more tissue samples. Among these, 92% had *Bartonella*-positive spleen samples, while 32% had positive ear samples. Blood was collected only from 10 deer in 2022, and 70% of *Bartonella*-positive deer from this cohort tested positive in one of the duplicate blood DNA samples. Notably, none of the deer had detectable *Bartonella* DNA in both duplicate blood samples. The percentage of positive samples for each tissue type in *Bartonella*-positive deer is illustrated in Appendix A.

Out of the total 470 samples, *Bartonella* DNA was detected in 245 (52%) using PCR targeting a 379 bp fragment of the *gltA* gene. Multiple infections involving more than one *Bartonella* genotype/species were common—sequencing confirmed them in 112 of the PCR-positive samples (46%). The remaining 133 sequences were analyzed, revealing a total of 23 different genotypes, all belonging to the ruminant-associated lineage 2.

### 3.5. Genotyping of Bartonella spp.

MLST was performed on a representative sample for each of the 23 genotypes found at the *gltA* locus (379 bp fragment), using the *gltA* (740 bp fragment), ITS, *nuoG*, and *rpoB* loci. Six genotypes had to be excluded from further analysis due to PCR failure or multiple infections identified through sequencing at other loci (these genotypes were each represented by only one or two samples). The *gltA* and ITS loci were very heterogeneous, with none of the 17 genotypes sharing 100% identity either with GenBank sequences or with another genotype from this study. A phylogenetic tree (Figure 1), constructed using concatenated sequences of all four loci, analyzed with Bayesian inference and maximum likelihood methods, revealed three distinct branches within lineage 2. Both algorithms placed ten genotypes within the *B. schoenbuchensis* complex (including three clustering with *B. capreoli*), six genotypes with *B.* sp. WD12.1 and *B*. sp. WD16.2, referred to as lineage B and lineage C in [47], and one genotype within *B. bovis*. *Bartonella schoenbuchensis* and *Bartonella* sp. clustering with lineage B and C were found in both deer and deer keds, but *B. bovis* (G08) was detected exclusively in tissue samples of deer.

Among the trees constructed using individual loci, the *gltA* tree (Appendix A) exhibited the closest similarity to the concatenated tree, identifying the same three branches. This locus also benefited from the highest availability of sequences in GenBank (182 total, including 55 unique sequences). Notably, a significant number of sequences were assigned to the lineage B and C branches (including lineages D and E from [47], obtained exclusively from *Lipoptena fortisetosa* or sika deer in Japan.

Phylogenetic trees for individual loci can be found in Appendix A, and detailed results for each sample in Appendix A.

### 3.6. Multiple Infections

For the most abundant genotype, G01, 17 samples were selected for MLST to investigate multiple infections and possible divergence at other loci. Among these 17 samples, 13 showed multiple infections at one or more loci. The majority of samples yielded sequences identical to a specific genotype group (G01–G17) across all loci (See Appendix A). However, none of the multiple infections allowed for reliable identification of all sequences within the mix.

### 3.7. Nanopore Sequencing of Multiple Infections

Due to the high prevalence of multiple infections observed in both the original *gltA* screening and subsequent MLST analysis, ONT sequencing was performed on selected samples. Specifically, 26 deer tissue samples exhibiting a mixed chromatogram in the 379 bp *gltA* screening were chosen. From these, high-quality sequences were successfully obtained for 22 samples (from 22 deer) following PCR amplification at the 740 bp *gltA* locus. In total, 72 high-quality contigs were generated, revealing 25 genotypes, including 14 (N01–N14) that had not been identified in the previous Sanger sequencing. All the sequences belonged to *Bartonella* lineage 2. The number of genotypes per sample ranged from 1 to 6, with a median of 3; the distribution is shown in Appendix A.

A genotype network was created using *gltA* sequences of genotypes G01–G17, ONT-detected genotypes N01–N14, and 10 reference genomes from GenBank (Figure 2).

## 4. Discussion

Significantly more *Lipoptena fortisetosa* (272; 84%) than *L. cervi* were collected in our study, indicating that *L. fortisetosa* is a well-established species in the area. Interestingly, the first known specimens of *L. fortisetosa* in Europe, dated back to 1933, were collected near Görlitz in Saxony, Germany [34], only 50 km from Krkonoše National Park, the sampling area of this study. The first published description of *L. fortisetosa* in Europe was based on keds collected near Opava, Czech Republic [69]. It is possible that *L. fortisetosa* has resided in central Europe for even longer but was previously misidentified as *L. cervi* due to their morphological and ecological similarities.

Phylogenetic analysis of *Lipoptena* COI revealed several haplotypes of *L. cervi* and *L. fortisetosa*. The heterogeneity observed in European samples (similar to that of *L. cervi*) suggests that the European *L. fortisetosa* population likely did not result from a single, limited invasion. Similar haplotypes are shared between Europe and northern continental Asia, with one isolate from Japan and two from Taiwan forming related branches.

A great diversity of *Bartonella* was discovered in the sample set: 17 unique MLST-defined genotypes belonging to at least three different species and 14 genotypes additionally detected by nanopore sequencing. This is all the more notable considering the small sampling area of Krkonoše National Park (350 km^2^). Additionally, all sequences at the gltA and ITS loci and most sequences at the rpoB and nuoG loci were unique compared to previously published ones, including isolates from recent studies in the nearby Saxony [44] and western Czech Republic [48].

Among the three deer tissue types used for PCR, spleen samples had the highest detection rate for *Bartonella*, followed by ear and blood samples. Three animals tested positive only in the ear samples, which may indicate an early-stage localized infection. Despite blood samples being collected in duplicates, no deer had detectable *Bartonella* DNA in both samples, suggesting low levels of bacteremia. This could also explain the low success rate of *Bartonella* cultivation from deer blood, as also reported in a previous study [29]. Even for symptomatic human patients, blood culture is negative in more than half of the cases [70].

Many PCR primers commonly used for the detection and genotyping of *Bartonella* in current studies were originally designed for *B. henselae* and may not align well with the DNA of other *Bartonella* species. This varied sensitivity must be considered when using these primers to detect *Bartonella* in environmental samples. To address this issue, modified or newly designed primers were used, achieving better results. Therefore, the use of such modified primers is recommended for detecting wildlife-associated *Bartonella* species. Among the 4 loci examined, *gltA* had the highest resolution, consistent with [44], and produced a phylogenetic tree most similar to the one based on concatenated sequences of all four loci.

Multiple infections were highly prevalent in both ked and deer samples, constituting nearly half of all initially obtained sequences. When MLST was performed on samples initially identified as identical at one locus, additional genotypes were found in the majority of cases. Nanopore sequencing of deer samples with multiple infections revealed up to six different genotypes co-occurring within a single sample. These findings underscore the widespread nature of *Bartonella* infection in deer. Despite this prevalence, the absence of detectable *Bartonella* DNA in some deer samples may be attributed to low bacterial loads. A recent study using the more sensitive qPCR method found a higher prevalence of 51% among red deer [29]. The high prevalence and frequency of multiple infections in cervids are likely linked to exposure to deer keds, as previously suggested [40].

All identified genotypes belonged to the ruminant *Bartonella* lineage 2 but encompassed various species: ten genotypes of *B. schoenbuchensis* sensu lato (three of *B. schoenbuchensis* subsp. *capreoli*), one of *B. bovis*, and six *Bartonella* sp. These six genotypes clustered with *Bartonella* sequences obtained from *L. fortisetosa* and sika deer in Japan, species not yet formally described and designed as lineages B to E [71]. Sika deer are native to East Asia but have been imported to Europe over the past two centuries, where they can hybridize with native red deer [72]. *Lipoptena fortisetosa* serves as a vector for *Bartonella* in sika deer in Japan [47], and it is hypothesized to have been introduced into Europe alongside sika deer or to have spread gradually eastward across Eurasia. Similar *Bartonella* genotypes have likely been found in Poland in both *L. cervi* [43] and *L. fortisetosa* [45]. Based on their geographical distribution, *Bartonella* lineages B–E may be associated with *L. fortisetosa*.

*Bartonella bovis* was detected in red deer for the first time, establishing them as the second known cervid host after moose [73]. In this study, similar to the findings of [73], *B. bovis* was exclusively found in deer samples and not in keds, suggesting that *L. cervi* and *L. fortisetosa* may not serve as vectors for this species. Conversely, in the USA, *B. bovis* DNA was detected in *Lipoptena mazamae* [74]. In contrast, *B. schoenbuchensis* subsp. *capreoli*, previously identified exclusively in roe deer and sika deer, was detected in several *Lipoptena fortisetosa* individuals.

## 5. Conclusions

The findings of this study demonstrate the unexpectedly huge *Bartonella* diversity in deer. The 17 detected genotypes were unique compared to previously published ones, even sequences from western Czech Republic and the neighboring Saxony, suggesting that the full richness of cervid *Bartonella* in central Europe has not yet been discovered. Nanopore sequencing discovered more genotypes and multiple infections, highlighting the high exposure of red deer to *Bartonella*, as well as the limitations of routine detection methods for low-bacteremia infections. Considering their ability to cause human illness and high prevalence, we should keep ruminant-associated bartonellae in mind as a sylvatic zoonotic pathogen. However, studies on human exposure to lineage 2 bartonellae and the vector potential of deer keds are needed to truly ascertain their significance.

## Figures and Tables

**Figure 1 pathogens-14-00006-f001:**
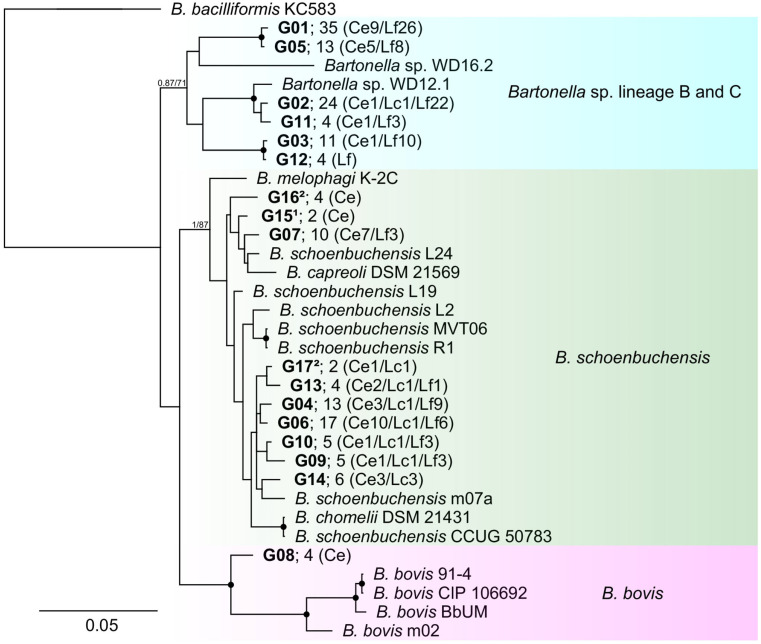
Phylogenetic tree of concatenated sequences. The tree was constructed using concatenated sequences of the *gltA*, *rpoB*, *nuoG*, and ITS loci of genotypes G01–G17 (in bold) and 10 genomes retrieved from GenBank. Bayesian inference was used for tree construction, with support values calculated using maximum likelihood with 500 bootstraps. Nodes with robust support (near 100%) in both models are indicated by black dots. Selected nodes are marked with numbers indicating Bayesian model support/bootstrap support. The number of individuals in which the genotype was detected is noted after the genotype name and the distribution among species in brackets (Ce = *Cervus elaphus*; Lc = *Lipoptena cervi*; Lf = *L. fortisetosa*). ^1^ Concatenate is missing the *nuoG* sequence. ^2^ Concatenate is missing the ITS sequence.

**Figure 2 pathogens-14-00006-f002:**
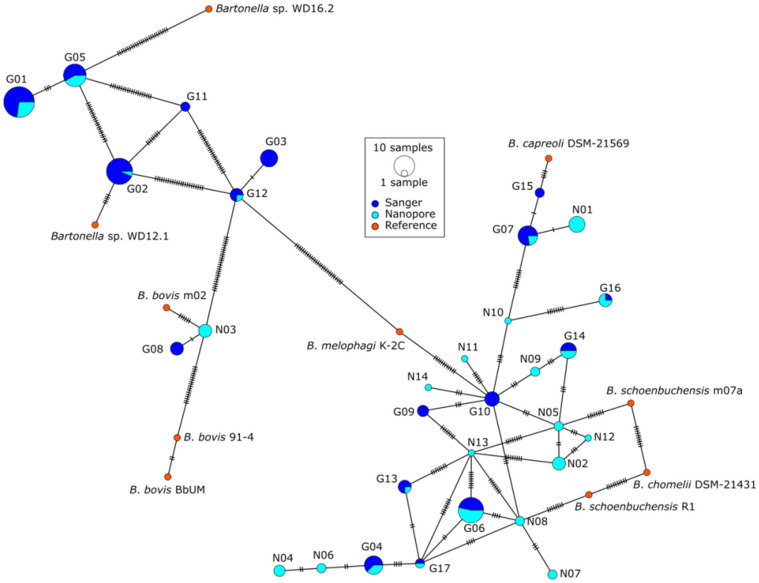
The genotype network of *gltA* sequences created using POPart. The sizes of the circles represent the number of deer groups (deer and all its keds) containing the genotype. Genotypes detected by Sanger sequencing are marked in dark blue, those detected by ONT in cyan, and reference genomes from GenBank in orange. Each notch on connecting lines represents one SNP.

**Table 1 pathogens-14-00006-t001:** List of primers used for the amplification of *Bartonella* DNA.

Locus	Reaction Name	Primer Name	Sequence 5′–3′	Source	Product Size (bp)
*gltA* 379 bp	gltA short	gltA_BhCS.781p	GGGGACCAGCTCATGGTGG	[54]	379
		gltA_BhCS.1137n	AATGCAAAAAGAACAGTAAACA	[54]	
*gltA* 740 bp	gltA-D	gltA_443F-D	GCYATGTCTGCATTYTATCA	[55] **	790
		gltA_1210R	GATCYTCAATCATTTCTTTCCA	[56]	
	gltA-D nested	gltA_443F-D	GCYATGTCTGCATTYTATCA	[55] **	740
		gltA_1137n	AATGCAAAAAGAACAGTAAACA	[54]	
ITS	ITS alt *	ITSalt_F	ATGATGATCCCAAGCCTTC	[57] **	700–1000
		ITSalt_R	CTTCTCTTCACAATTTCAATAGAAC	This study	
*nuoG*	nuoG *	nuoG_F	GGCGTGATTGTTCTCGTTA	[58]	366
		nuoG_R	CACGACCACGGCTATCAAT	[58]	
*rpoB*	rpoB-D	rpoB_1400F-D	CGCATTGGYTTRCTTCGTATG	[59] **	893
		rpoB_2300R-D	GTAGAYTGATTRGAACGCTG	[59] **	
	rpoB-D nested	rpoB_1596F-D	CGCATTATGGTCGTATTTGTCC	[59] **	628
		rpoB_2300R-D	GTAGAYTGATTRGAACGCTG	[59] **	

* repeat reaction with first product if needed; ** slightly modified from the original sequence.

**Table 2 pathogens-14-00006-t002:** The number of samples collected each year. Spleen and ear samples were only collected in 2020–2022, and blood samples were only collected in 2022. For keds, Lc = *L. cervi* and Lf = *L. fortisetosa*.

Year	2016	2020	2021	2022
Deer examined	38	47	57	10
Keds collected	129 Lc:15, Lf:114	99 Lc:11, Lf:88	85 Lc:23, Lf:62	9 Lc:1, Lf:8
Spleen samples		46	57	10
Ear samples		47	57	10
Blood samples				10

## Data Availability

Sequences were uploaded to Genbank under the accession numbers PP529989–PP530038, PP531479–PP531493, PP940817–PP940820 and PP940824–PP940827.

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
