# Peer review of "Diversity and Multiple Infections of Bartonella in Red Deer and Deer Keds"

_pathogens, 2024, doi:10.3390/pathogens14010006_

Round 1
Reviewer 1 Report
Comments and Suggestions for Authors
1. Line 16: Italicize Bartonella here and at all other occurrences.
2. Line 17: What is `wildlife bartonellae`? Shouldn't it be wildtype? Also, is it Bartonella or Bartonellae?
3. Line 65-66: Why is there a superscript `15` at the end of this sentence?
4. Line 70: What is sensu lato?
5. Line 75: Consider including a paragraph in introduction that guides readers to what the gaps are in present research and what this work is addressing.
6. Line 85-86: How long was the incubation for?
7. Line 186: How did one blood sample test positive when all 10 blood samples could not be cultivated?
8. Line 82: How are authors analyzing blood samples when they could not be cultivated? Authors mention 10 blood samples were used in 2022 and could not be cultivated.
Additionally, I would also suggest writing a more comprehensive discussion section that includes different samples analyzed and their significance.
Author Response
Thank you for the fair review. I have taken your comments into account and made the following changes:
1. Line 16: Italicize Bartonella here and at all other occurrences.
Fixed throughout the manuscript and in reference list.
2. Line 17: What is `wildlife bartonellae`? Shouldn't it be wildtype? Also, is it Bartonella or Bartonellae?
Changed to wildlife-associated bartonellae to be clearer (L17). The plural bartonellae is used to emphasize it is multiple species.
3. Line 65-66: Why is there a superscript `15` at the end of this sentence?
An error in the citation format. Fixed, L70.
4. Line 70: What is sensu lato?
Sensu lato (“in a broad sense”) is used in taxonomy to refer to a taxon that was previously thought to be several species, but was then unified. According to the convincing analysis in do Amaral et al. (2022), this is the case for B. schoenbuchensis. B. melophagi, B. capreoli and B. chomelii should be subspecies of B. schoenbuchensis (B. schoenbuchensis sensu lato), while the species previously described as B. schoenbuchensis becomes B. schoenbuchensis sensu stricto (“in a strict sense”). Another possible phrasing is B. schoenbuchensis complex.
5. Line 75: Consider including a paragraph in introduction that guides readers to what the gaps are in present research and what this work is addressing.
Good point. Added in L79-83.
6. Line 85-86: How long was the incubation for?
Four weeks. Added into the text, as well as a citation for the methodology, L95-99.
7. Line 186: How did one blood sample test positive when all 10 blood samples could not be cultivated?
Cultivation of Bartonella from blood of large mammals is rarely successful even in infected individuals. A comment about this has been added to the discussion in L296-299.
8. Line 82: How are authors analyzing blood samples when they could not be cultivated? Authors mention 10 blood samples were used in 2022 and could not be cultivated.
Along with attempted cultivation, the samples were analyzed by PCR like all others.
Additionally, I would also suggest writing a more comprehensive discussion section that includes different samples analyzed and their significance.
The discussion of different sample types is present in L292-294. For a concise conclusion of the findings, as well as future perspectives, a Conclusion section has been added at the end (L338-347).
Reviewer 2 Report
Comments and Suggestions for Authors
The manuscript provides useful information in the knowledge of the molecular epidemiology of Bartonella spp. in red deer and deer kids within a central European country. I support its further processing after some clarifications as outlined below:
L14: “we tested” – please avoid the personal mode verb formulations, may sound unprofessional in scientific papers. Please revise this concern throughout the manuscript
L18: “Bartonella” – please ensure the italics writing of the scientific names of species throughout the manuscript, including the reference list (e.g. B. schoenbuchensis in L20, etc.)
L29: “… including humans” – please ensure appropriate citations after the end of each statement. The first mentioned reference is only in line 38. Please carefully revise this issue throughout the Introduction section.
L38: While the article mentions the zoonotic potential of Bartonella spp. there is a lack of references to global epidemiological data, especially regarding its prevalence in humans. Including relevant studies or data on human cases would strengthen the argument about the zoonotic potential of this pathogen.
L38-39: Some phrases are too informal for a scientific article, such as "can pose risk to hunters". A more formal construction, such as "might increase the risk to hunters" would be more appropriate.
L41-43: the sentence mentioned within these lines is unclear, how to pose a public health risk the “forest visiting”? Please provide details.
L46: “B. dromedarii and B. melophagi” – please provide appropriate references for these species
L48: “ANI (average nucleotide identity)” – must be “average nucleotide identity (ANI)”
L67: “keds 15.” – please use uniform citations throughout the manuscript
L77: the sampling methodology is unclear, please provide details in sample size calculations, quantities, preservation, transport, etc.
L84: the Bartonella cultivation is poorly described – please provide details and mention appropriate references
L85: “SIMB medium” – for each of the used reagents and devices please uniformly provide throughout the manuscript the production company name, city, and country.
L95: “-20°C” –please replace the hyphen with a minus sign. Revise throughout the manuscript.
L102: it is unclear if the authors used positive and negative controls during PCR reactions in order to validate their results. In addition, was an internal control used in the PCR reaction for this study? If so, please include this information in the PCR analysis section.
L134: the phylogenetic analysis section must be completely restructured in a more “reader-friendly way”, meaning the removal of some technical details
L170: Table 2 It would be helpful to add a column showing the hunting ground locations, allowing for quicker identification of geographical patterns.
I would advise the authors to provide a separate Conclusion section that presents the key results and indicates future research perspectives.
Author Response
The manuscript provides useful information in the knowledge of the molecular epidemiology of Bartonella spp. in red deer and deer kids within a central European country. I support its further processing after some clarifications as outlined below:
L14: “we tested” – please avoid the personal mode verb formulations, may sound unprofessional in scientific papers. Please revise this concern throughout the manuscript
Edited in lines 15, 37, 250, 285, 304, 305, 335, 338.
L18: “Bartonella” – please ensure the italics writing of the scientific names of species throughout the manuscript, including the reference list (e.g. B. schoenbuchensis in L20, etc.)
Fixed throughout manuscript and references.
L29: “… including humans” – please ensure appropriate citations after the end of each statement. The first mentioned reference is only in line 38. Please carefully revise this issue throughout the Introduction section.
Added citations for B. henselae, B. quintana as well as other species known to infect humans, L29-32. Added a citation for LPSN in L47.
L38: While the article mentions the zoonotic potential of Bartonella spp. there is a lack of references to global epidemiological data, especially regarding its prevalence in humans. Including relevant studies or data on human cases would strengthen the argument about the zoonotic potential of this pathogen.
This is true. Unfortunately, there is no comprehensive epidemiological data about zoonotic Bartonella (apart from B. henselae). In clinical practice, bartonellosis is diagnosed through serological testing, which doesn’t allow for definite species determination. Diagnosis with PCR or cultivation is very rare. Because of this, reports of different Bartonella spp. in humans are limited to the individual cases cited in the article. The section in L53-58 has been rephrased to emphasize this.
L38-39: Some phrases are too informal for a scientific article, such as "can pose risk to hunters". A more formal construction, such as "might increase the risk to hunters" would be more appropriate.
“can pose” was changed to “might pose” in line 40.
L41-43: the sentence mentioned within these lines is unclear, how to pose a public health risk the “forest visiting”? Please provide details.
The risk may come from encountering deer keds, which are known to land on humans and scratch/bite them, as explained in L66-71.
L46: “B. dromedarii and B. melophagi” – please provide appropriate references for these species
Fixed, L47-48.
L48: “ANI (average nucleotide identity)” – must be “average nucleotide identity (ANI)”
Fixed, L49.
L67: “keds 15.” – please use uniform citations throughout the manuscript
Fixed, L70.
L77: the sampling methodology is unclear, please provide details in sample size calculations, quantities, preservation, transport, etc.
More information about sample handling and quantities has been added in L88-93 . The samples were obtained from animals shot during routine hunting by collaborating hunters. As such, we could not set a sample size in advance, and just processed all samples the hunters provided us during the four-year collaboration.
L84: the Bartonella cultivation is poorly described – please provide details and mention appropriate references
The section was rephrased, and a reference outlining the followed recommendations was added, L95-99.
L85: “SIMB medium” – for each of the used reagents and devices please uniformly provide throughout the manuscript the production company name, city, and country.
Added.
L95: “-20°C” –please replace the hyphen with a minus sign. Revise throughout the manuscript.
Fixed the minus sign, as well as en-dashes throughout the document.
L102: it is unclear if the authors used positive and negative controls during PCR reactions in order to validate their results. In addition, was an internal control used in the PCR reaction for this study? If so, please include this information in the PCR analysis section.
Yes, positive and negative controls were used. Added in L130-131.
L134: the phylogenetic analysis section must be completely restructured in a more “reader-friendly way”, meaning the removal of some technical details
Some technical details were removed (L172-173) and the section has been slightly rephrased to shorten it.
L170: Table 2 It would be helpful to add a column showing the hunting ground locations, allowing for quicker identification of geographical patterns.
Unfortunately, the exact locations of hunted deer are not available. Each year they were hunted in multiple locations in the Krkonoše National Park. For a better sense of scale, the area of the park was added in L87.
I would advise the authors to provide a separate Conclusion section that presents the key results and indicates future research perspectives.
Thank you for the suggestion, as well as your fair review. A Conclusion section has been added, using some lines from the end of the Discussion section, as well as new text (L338-347)
Round 2
Reviewer 1 Report
Comments and Suggestions for Authors
No more comments.
Reviewer 2 Report
Comments and Suggestions for Authors
The authors have correctly acknowledged all of my raised concerns.